# Parental Genetics Communicate with Intrauterine Environment to Reprogram Newborn Telomeres and Immunity

**DOI:** 10.3390/cells11233777

**Published:** 2022-11-25

**Authors:** Sadia Farrukh, Saeeda Baig, Rubina Hussain, Rehan Imad, Maria Khalid

**Affiliations:** 1Department Biochemistry, Ziauddin University, Karachi 74600, Pakistan; 2Department Gynecology and Obstetrics, Ziauddin University, Karachi 74600, Pakistan; 3Department Molecular Medicine, Ziauddin University, Karachi 74600, Pakistan

**Keywords:** telomere, telomerase, TERT, polymorphism, telomere length (TL), DNA repair, reprogramming, immunity

## Abstract

Telomeres, markers for cellular senescence, have been found substantially influenced by parental inheritance. It is well known that genomic stability is preserved by the DNA repair mechanism through telomerase. This study aimed to determine the association between parents–newborn telomere length (TL) and telomerase gene (TERT), highlighting DNA repair combined with TL/TERT polymorphism and immunosenescence of the triad. The mother–father–newborn triad blood samples (*n* = 312) were collected from Ziauddin Hospitals, Pakistan, between September 2021 and June 2022. The telomere length (T/S ratio) was quantified by qPCR, polymorphism was identified by Sanger sequencing, and immunosenescence by flow cytometry. The linear regression was applied to TL and gene association. The newborns had longest TL (2.51 ± 2.87) and strong positive association (R = 0.25, *p* ≤ 0.0001) (transgenerational health effects) with mothers’ TL (1.6 ± 2.00). Maternal demographics—socioeconomic status, education, and occupation—showed significant effects on TL of newborns (*p* < 0.015, 0.034, 0.04, respectively). The TERT risk genotype CC (rs2736100) was predominant in the triad (0.6, 0.5, 0.65, respectively) with a strong positive association with newborn TL (β = 2.91, <0.0011). Further analysis highlighted the expression of KLRG 1+ in T-cells with shorter TL but less frequent among newborns. The study concludes that TERT, parental TL, antenatal maternal health, and immunity have a significantly positive effect on the repair of newborn TL.

## 1. Introduction

Telomeres are nucleoprotein structures present at the end of the chromosomes, formed from noncoding tandem repeats TTAGGG. They play an important role in maintaining genomic stability by DNA repair mechanism and protect chromosomes from end-to-end fusion (dicentric chromosome) and degradation [1]. Telomeres progressively shorten because of the incapability of DNA polymerase due to discontinuous elongation on lagging strands during each cell division. Telomeres also prevent exonucleolytic degradation, ligation, and inappropriate DNA repair reactions. The critically short telomere length triggers the cell to enter replicative senescence, which subsequently leads to cell death [1]. Therefore, telomere length (TL) is an important determinant of telomere function and a biomarker of biological aging as well as the development and progression of diseases [2]. An emerging problem of telomeropathies is due to compromised telomere structure and function resulting from DNA damage and shelterin protein disruption, also named “telomere disorders,” or “telomere syndromes” [3].

The TL is longer in newborns due to the action of telomerase in the intrauterine embryonic period, but decreases with age causing a three to eight times increased risk of mortality and morbidity in older people [4]. Therefore, the high activity of telomerase during pregnancy may protect fetal telomeres against some intrauterine lesions. However, during the intrauterine period, maternally exposed environmental parameters such as physical activity, education, and stress can disturb the health of the growing child due to the genetic programming of a fetus [5,6]. Such modifications can alter TL, which may lead to the complication of small or large gestational age (SGA/LGA), pre-eclampsia, intrauterine growth restriction (IUGR), oxidative stress, and hypoxia [7,8,9,10].

The average leukocyte telomere length may be used as an indicator of a biological or surrogate marker that can determine an individual’s TL [11]. The newborn TL is related to many risk factors, such as gender, body mass index (BMI), hormone (higher estrogen), physical inactivity, smoking, stress, paternal age, and socioeconomic status (SES) [6,12,13,14].

Telomerase, a DNA reverse transcriptase polymerase, is involved in TL maintenance. It consists of the major TERT gene, located on chromosome 5p15.3, consisting of 16 exons and 15 introns spanning 42 kilobases, and uses an RNA template to add telomeric DNA, thus compensating for the telomere shortening caused by cell divisions [15]. They are present in stem, embryonic, and cancer cells, but not in somatic cells except in special situations, such as during tissue repair and tumors. Telomere may also act as a “tumor suppressor system” because it limits the proliferation of transformed cells. It extends the 3′ DNA end rather than an RNA primer [16]. Mutation or genetic variation or polymorphism in TERT can cause aberrant telomerase activation that can be a reason for uncontrolled cell division, the breakdown of the division cycle, and the repair of damaged DNA, which may lead to oncogenesis [17].

On the other hand, under different physiological or pathological conditions, the exposure of cells to various stress signals activates the p53 signaling pathway together with several transcriptional programs, including cell cycle arrest, DNA repair, senescence, and apoptosis, hence leading to the suppression of tumor growth [18]. However, nondividing, senescent cells due to terminal differentiation might accumulate with aging. Such processes are caused by a progressive shortening of telomeres. The senescent T-lymphocytes include increased expression of killer cell lectin-like receptor G1 (KLRG1) and CD57. The KLRG1 is an inhibitory C-type lectin-like receptor, a transmembrane glycoprotein with an extracellular domain homologous to C-type lectins having a cytoplasmic tail containing an immunoreceptor tyrosine-based inhibition motive (ITIM). KLRG1 is expressed in T cells and NK cells [19]. The other glycoprotein CD57, the human natural killer-1 (HNK-1), is found on many NK cells but also a subset of CD8 cells. It has been found to identify terminally differentiated T cells with reduced proliferative capacity. Only CD57+ KLRG1+ double-positive T cells can compromise their ability to proliferate [20].

Based on the above facts, the rationale of this study was that there is a scarcity of data regarding parental TL (especially fathers) and its association with various risk factors affecting newborn telomeres and immunosenescence and long-term impact on health. Moreover, in the targeted region (Karachi, Pakistan) TERT gene single nucleotide polymorphism (SNP) and minor allele frequency (MAF) has not been to date explored. In our previous study [21], we proved that mothers had shorter telomere lengths compared to their newborns, thus highlighting fetal telomere reprogramming or repair. The current study may add to the data by potentially highlighting the multiple risk factors affecting telomeres and telomerase gene SNP of mother and father along with immune senescence and their effect on newborns’ genetics. Such a discovery will be significant for the identification of inheritance patterns in newborns and highlighting risk factors involved in TL variation that may act as a biological marker for risk prediction of different diseases.

Therefore, this study aimed to provide a comprehensive analysis of the mother and father TL, demographics of parents, and different maternal clinical risk factors and immune senescence effect on the newborn TL (outcome). Sex-specific effects and the contributions of factors to variation in newborn TL were also assessed. TERT gene polymorphism and its association with TL, along with the frequency of genotype and allele, were also investigated.

## 2. Materials and Methods

This cross-sectional study recruited *n* = 312, parents–newborn (triad) (104/each), from Ziauddin Hospitals, after the ethics approval from the Ethics Review Committee (Ref. No. 3950721SFBC) of Ziauddin University. The samples were collected after taking informed consent from both parents between September 2021 and June 2022. All of the recruited females were aged 18–35 years with a gestational age of 35 weeks or above based on ultrasound data. Males aged 18–45 years were included. Maternal and offspring assessments were recorded, or data were obtained from hospital records. Females and males with any known malignancies were not included in the study. Tobacco users (smoker, chewer) were separated during data collection and presented as a comparison to nonsmokers. Drug-addicted (opium, hashish, etc.) pregnant females or males were also excluded. The questionnaire was filled out to gather information on parental age, socioeconomic status (SES), ethnicity, occupation, and education. The SES was measured by considering income as a major variable [22], considering low: <20,000, lower middle: 20,000–35,000, upper middle: 36,000–100,000, and high: >100,000 (dollar rate 1 May 2022) according to the World Bank data [23]. Maternal features such as parity, miscarriage, gravidity, and laboratory investigations (hemoglobin, total leukocyte count, RBS, and platelets) were noted. The newborn parameters included mode of delivery (MOD), birth weight, gestational age, gender, heart rate, respiratory rate, occipital frontal circumference (OFC), and newborn length, which were recorded after birth.

### 2.1. DNA Extraction from the Blood

Five milliliters of venous blood of females before delivery and 5 mL of the blood of males were collected in ethylenediaminetetraacetic acid (EDTA) tubes after taking written consent from both females and males. Then, 5 mL of umbilical venous cord blood was collected into EDTA tubes immediately after delivery when the cord was still in contact with the placenta. All the collected blood was stored at 4 °C until DNA extraction.

DNA was extracted from the blood by Qiagen DNA Blood Mini Kit (catalog no. 51306, Hilden, Germany) method according to the company protocol. After extraction, the DNA samples were stored at −80 °C. The concentration of each DNA sample was measured by using a Multiskan Sky spectrophotometer (Thermo Fisher Scientific, Waltham, MA, USA) at 260 and 280 nm, and a 260/280 ratio was noted.

### 2.2. QPCR for Telomere Length (TL) Quantification

For telomere length quantification, leukocyte telomere length (LTL) was used in qPCR following the Cawthon et al. multiplex method [24]. The reference DNA (pool of four healthy males and females) was used as a standard in all runs of qPCR. The 5 dilutions (150–1.85 ng) of reference DNA were used as a standard to form the standard curve. Then, the experimental DNA of the mother, father, and cord was quantified by qPCR. The qPCR reaction included 15 µL PCR Master Mix that contained 11 µL of PCR Maxima syber green master mix (SYBR Green 1 Dye, AmpliTaq Gold™ DNA Polymerase, dNTPs with dUTP and ROX reference dye) (catalog no. K0221 Thermo Fisher Scientific) and 1 µL of forward and reverse primer (1 μM) of telomere and human Betaglobin gene (single copy gene). Then, for each reaction, a 10 µL of DNA sample was added to individually labeled PCR tube, bringing the final volume of PCR reaction Mix to 25 µL.

The thermal cycler (Agilent, Santa Clara County, CA, USA) was programmed according to the recommendations in [24]. Each experimental sample was analyzed in triplicate. The T/S ratio was calculated by dividing the nanograms of the standard DNA that matched the experimental sample for the copy number of the telomere template (T) by the number of nanograms of the standard DNA that matched the copy number of the human beta-globin (single copy gene) (S). A wide range of input DNA amounts was used, as long as the signals of both T and S fell within the standard curve range. Because each experimental sample was assayed in triplicate, the average of the three T/S values was used as the average telomere length per cell. Samples with a T/S > 1.0 had a greater average TL than that of the standard DNA and, similarly, T/S < 1.0 had a smaller TL than standard DNA.

### 2.3. Sanger Sequencing of TERT Gene Polymorphism

A region of the TERT gene rs2736100 SNP was selected based on its association with TL and had a minor allele frequency (MAF) greater than 5% in the Pakistani population identified from a 1000 genome database. The gene locus was amplified by PCR using forward primer: 5′CTCGGAGCCTCATCCTTTGT3′ and reverse primer: 5′TCTCAGGCATCTTGACACCC3′ (Synbio Tech, Monmouth Junction, NJ, USA), prepared by Primer 3 software. The total volume of PCR reaction mixture used was 50 μL: 25 μL of DreamTaq master mix (catalog no. K0181, Thermo Fisher Scientific), 2 μL of 1 μM forward primers, 2 μL of 1 μM reverse primers, 10 μL of 10–50 ng DNA template, and 11 μL water. The selected primer amplified 544 base pairs for the TERT region. The thermal cycling conditions were as follows: initial denaturation (1 cycle) at 95 °C for 3 min, then 40 cycles of denaturation at 95 °C for 30 s, annealing at 57 °C for 30 s, extension for 72 °C for 1 min and 1 cycle of final extension at 72 °C for 15 min. No template control (NTC) reactions were also run in all reactions. The PCR products were then purified by ExoSap-IT PCR Product Cleanup kit according to the protocol (catalog no. 78200, Thermo Fisher Scientific). The sequence analysis (*R*^2^ = 0.90) was performed by using a Big Dye Terminator v.3.1 Sequencing Kit (catalog no. 4337456, Thermo Fisher Scientific) on SeqStudio Genetic Analyzer (Thermo Fisher Scientific).

### 2.4. Flow Cytometry for Immunosenescence Detection

For cryopreservation, blood was collected in EDTA tubes from each participant and processed within 24–48 hours of collection. The cryopreservation was done by the procedure already mentioned [25]. The cells were first kept at −20 °C for 1–2 hours and then preserved at −80° C for long-term storage. For analysis, PBMCs were resuspended in media (fetal bovine serum (FBS) 10% and cell culture media 90%) and centrifuged for 10 min at 1600 rpm. Then the supernatant was discarded and the pellet was washed with phosphate buffer saline (PBS) and again centrifuged for 10 min at 1600 rpm. Cells were then suspended in 1 mL PBS and then labeled with 1 µL of antibodies: PerCP labeled CD45 Monoclonal Antibody (HI30) (catalog no. MHCD4531, Thermo Fischer Scientific), PE labeled KLRG1 Monoclonal Antibody (13F12F2), (catalog no. 12-9488-42, PE, eBioscienceTM, Thermo Fischer Scientific), and FITC-labeled CD57 Monoclonal Antibody (TB01 (TBO1)) (catalog no. 11-0577-42CD57, eBioscienceTM, Thermo Fisher Scientific). Cells were then incubated for 30 min at 4 °C in the dark and were immediately analyzed using a flow cytometer (FACS caliber, BDe Bioscience) [26]. Lymphocytes were gated by using CD45 and analyzed for immunosenescence by KLRG1 and CD57 expression. Data were acquired and subsequently analyzed by using the software.

### 2.5. Statistical Analysis

Data were analyzed by using Statistical Package for Social Sciences (SPSS) version 24. The quantitative variables were calculated as mean ± standard deviation (SD), and qualitative variables were measured as frequency and percentages. For categorical, i.e., ethnicity, smoking status, gravidity, parity, miscarriages, occupation, immune markers CD57+, KLRG+, and ordinal, i.e., education and socioeconomic status predictors, the Kruskal Wallis test was used. The Mann–Whitney U test was also used to check the comparison between genders of newborns. A simple linear regression analysis was used for the association of paternal TL (T/S ratio) with newborn TL (T/S ratio) (outcome) at a 95% confidence level. The graphs were prepared by GraphPad Prism Software. Mega x software was used for the analysis of sequencing and identification of minor alleles. The SHESIS software (http://analysis.bio-x.cn/myAnalysis.php, accessed on 15 July 2022) to analyze the genotype and allele frequency distribution. The threshold for statistical significance was defined as *p* < 0.05.

## 3. Results

### 3.1. Characteristics of Participants and Telomere Length (TL) Variation

The demographics of *n* = 312, parents–newborn (triad) (104/each) and their telomere length (T/S ratio) are presented in Table 1.

The mean (mean ± SD) age of the mothers and fathers was 27 ± 3.94 and 34 ± 6.01, respectively. The mean TL of a mother was 1.6 ± 2.0, which was greater than the father’s TL of 1.49 ± 1.63; however, newborns showed the longest TL of 2.51 ± 2.87 among all (Figure 1). A highly significant association was found between mother TL and newborn TL (R = 0.25, *p* ≤ 0.0001). The father TL and its association with newborn TL were also found significant (R = 0.09, *p* = 0.0019). The comparison between mother and father TL also gave significant results (*p* = 0.023) by linear regression (R = 0.05).

Socioeconomic status (SES) showed significant results among the triad. The longest TL was seen in mothers and newborns of upper-middle SES (1.99 ± 2.71, 2.58 ± 3.86) and fathers of low SES (1.83 ± 1.84). Nonworking females had long TL (1.70 ± 2.03), but their newborns had small TL (2.23 ± 2.3) (*p* = 0.034). However, the unemployed fathers were found with smaller TL (1.26 ± 0.70). Among all levels of education, significant results (*p* = 0.03, 0.04) were seen in graduate mothers and their newborns with the longest TL (2.17 ± 1.67, 2.78 ± 2.89), whereas fathers having less than high school education showed the longest TL (1.94 ± 1.80). The ethnicity did not add any significant result (<0.05) to the study; however, it was observed that Punjabi ethnicity had the longest TL in mothers and fathers (2.83 ± 2.80, 1.89 ± 2.49), but newborns of Pathan had the longest TL (2.73 ± 3.02). It was discovered that smoker mothers and their newborns had shorter TL (1.64 ± 1.61, 2.08 + 1.98), but their fathers had long TL compared to the nonsmoker group (2.13± 2.08) (Table 1).

Mothers with an age range of 21–25 showed the longest TL in newborns; whereas, fathers with age > 35 showed the longest TL in newborns, and this increased with an increase in the father’s age. On the other hand, in comparison to parents’ age and effect on TL, a decreasing pattern was seen with the increase in age (Figure 2A–D).

### 3.2. Antenatal Clinical Risk Factors and TL Variations in Newborn

Clinical conditions and TL measurement summary of all female participants in this study are shown in Table 2, indicating maternal TL and its effect on newborn TL. Mothers with some pathological conditions (gestational diabetes mellitus (GDM), preeclampsia, anemia, and thalassemia) and their newborns had smaller TL (1.38 ± 1.90, 2.49 ± 2.97) (*p* = 0.000) compared to the healthy mothers.

The longest TL (1.94 ± 3.03) (*p* = 0.05) was seen in females with Hb greater than 12 mg/dl hemoglobin level. The total leukocyte count (TLC) range 8.1–10 had highest frequency of 36 (35%) and >12 10^9^/L TLC in patients that had the longest TL (1.92 ± 2.78) (*p* = 0.09). The primigravida 39 (37%) had longer TL (1.34 ± 1.72) compared to multiparous females. The newborn to females with no parity had the longest TL (2.75 ± 2.16) (*p* = 0.025) compared to multiparous females (1.80 ± 3.12). Similarly, in parameter gravidity, the TL of females decreases with the increase in gravida. On the contrary, newborn TL decreased with increased gravida as seen in parity (Table 2).

### 3.3. Newborn’s Characteristics and Their Effect on TL

The newborn frequency showed 54% girls and 50% boys. The girls average TL at birth was longer (2.72 ± 3.09) than boys (2.13 ± 2.39) (*p* = 0.000) (Figure 3).

Newborn TL was not associated with gestational age and birth weight (Figure 4A,B). However, the gestational age and newborn weight had positive association (R = 0.21, *p* = 0.001). The lower (uterine) segment caesarean section (LSCS) (56%) was a more common mode of delivery compared to spontaneous vaginal delivery (SVD) (54%). The difference between gestational age and TL was not found to be significant (*p* = 0.20); however, the more the gestational age (36–38), the longer the TL, and vice versa (Table 3).

Regarding birth weight, most boys and girls were in the range 2.6–3.0 kg 31 (61%), 26 (51%)), but the longest TL was found in newborns with 3.1–3.5 kg weight (3.60 ± 2.31, 2.83 ± 3.17). The heart rate (HR) was found significant (*p* = 0.03) within the range of 130–140 and respiratory rate (RR) (beats per minute) was between 51 and 60 in both genders. The 40–50 cm length was observed mostly in girls 38 (70%) and boys 35 (70%), highlighting different TL (2.94 ± 3.29 and 1.92 ± 1.95). The OFC (cm) was seen between 30 and 35 among girls 48 (89%) and in all boys (*p* = 0.01) (Table 3). A comparison of the paternal and maternal TL effects showed no association (R = 0.005 *p* = 0.58, R = 0.03 *p* = 0.18) with either girls or boys in the same model, although girls’ TL had a longer length (Figure 5).

### 3.4. The Heritability of TL Influenced by Telomerase (TERT) Genetics

The influence of genetics on the heritability of TL from either mother or father was examined via performing Sanger sequencing (*R*^2^ = 0.90). Two different alleles (C and A) and three genotypes (AC, AA, CC) of TERT gene were identified. The heterozygous genotype, CC, was found dominant in parents (0.6, 0.5) and their newborn (0.65), whereas the homozygous genotype AA (0.3) was only found in fathers (Figure 6, Table 4).

On further regression analysis of TERT rs2736100 SNP had a significant association among genotypes and TL (*p* < 0.01) in the triad, but newborn TL showed a strong positive association (β = 2.91, *p* < 0.0011) (Table 4). However, the longest TL (1.35 ± 1.21, 1.40 ± 0.13, 3.47 ± 3.7) was seen in AC genotype according to the length of the PCR fragment. The minor allele frequency A was less frequent (0.2) and had a shorter TL (*p* = 0.00).

### 3.5. Immune Senescence Markers CD57 and KLRG1 Expression

When immunosenescence and telomere length was compared, the CD45 cell surface marker revealed a significantly higher proportion of CD57− KLRG1− cells observed in the newborns compared to the fathers and mothers (Figure 7). Whereas in the father there was a significantly(p < 0.05) higher percentage of CD57+ KLRG1+ expression, KLRG1+ cells were significantly lower in newborns. The telomere length was found longest in CD57− KLRG1− of newborns and shorter in the KLRG1+ cells of fathers and mothers (Figure 8).

## 4. Discussion

To the best of our knowledge, this study is the first to investigate newborn TL and its association with parental telomere genetics and immune senescence in the subset of Karachi, Pakistan. In this study, we identified significant factors influencing newborn TL: age, SES, education, antenatal health of females, genetic variation in the TERT gene, and immunosenescence markers.

The longest TL was found in the newborns, overall in the mother–father–newborn triad. The above analysis was further confirmed by mothers with longer TL giving birth to a newborn with longer TL with a positive association, thus emphasizing fetal programming or repair of the telomere biology that may affect the health and lifespan of offspring [27].

The length of the telomere was found associated with the *TERT* gene (rs2736100), targeting C/A alleles. Shorter TL was related to *TERT* variant’s minor allele A in the targeted population compared to the allele C as the major allele, which is also supported in a study by Cui et al. [28]. Genetic variation or polymorphism of the TERT gene acts as an important factor well known for the heritable effects on TL. *TERT* encodes a major protein of the enzyme telomerase required for telomeres synthesis, maintenance, and repair. Whereas, in different studies (Table 5), C/T are considered common alleles and have an association with both TL and aging [15,18,29]. However, another maternal–newborn study from China observed T allele association with shorter TL and found its effective allele frequency (EAF) to be 0.570 in mothers and 0.594 in the cord (newborn) (Table 5) [27].

Molecular disruptions, such as mutation or genetic variation in both TERT along with p53 gene, can alter expression and often lead to aberrant telomerase activation that can induce uncontrolled cell proliferation and miss the DNA repair process, causing oncogenesis [32]. A meta-analysis highlighted that the *TERT* SNP rs2736100 of C allele is associated with multiple cancerous diseases, whereas the A allele is associated with a predisposition to especially degenerative noncancerous diseases [33]. In addition to previous research, another study confirmed that AC polymorphism was a strong risk factor for idiopathic pulmonary fibrosis (IPF) but not for interstitial lung diseases; this supports the results of the current study by highlighting that individuals with AC genotype are at risk of developing diseases [28].

This study, for the first time reported in the literature, measured the changes in telomere length of lymphocyte (T-cell) subsets in mothers, fathers, and newborns. The telomeres were found shorter in cells with more expression of KLRG1+ than CD57+KLRG1+. The cells that had the shortest TL have been used extensively as a marker of biological age because they erode progressively with each round of cell division, thus critically shortened telomeres trigger mechanisms for senescence, causing the cell to undergo proliferative arrest [34]. However, the cells with the expression of the immune marker KLRG1 were found less frequent in newborns than adult cells of fathers and mothers, as already explored in fetal and adults by Schreurs et al. [35]. The T-cells that have reached terminal differentiation are known to have exceedingly strong telomere erosion and lead to critically short telomeres, which results in cellular senescence or apoptosis. Therefore, cells in fathers and mothers of targeted populations with high expression of immune aging markers could potentially destabilize cell genomics [36]. Such cellular senescence at an early stage causes perturbations of cell growth and DNA repair dysregulation, initiating senescent cell accumulation, which is the turning point for disease initiation or malignancy.

Maternal and newborn TL showed significant results with maternal age (18 to 35 years) but not with parental age (18–45). However, the findings also showed that newborns of older fathers had longer TL in the analysis. Previous studies have also reported an increase in sperm TL (TL plasticity increases with age and is higher in sperms than in oocytes), which may be a potential reason for the newborn to inherit longer telomeres in older fathers. The increase in sperm TL and its underlying biological mechanisms are less known; however, selective loss of spermatogonia with short TL due to apoptosis and an increase in proliferation of spermatogonia with high testicular telomerase activity causes the TL to be longer [37,38,39].

Generally, there is a gap in the literature concerning the association between parents and their newborns’ TL. In this study, parents/newborn TL associations were studied regarding demographics, including socioeconomic status (SES) and health disparities. The results were statistically significant (*p* < 0.05) between newborn TL and parents’ SES among all the subgroups (low, lower middle, upper middle, high). TL of mothers and their newborns from upper-middle SES were found longer, compared to the other SES, which is in accordance with our previous study [21]. Previously, mixed results were observed on SES and telomere length [40,41]. However, shorter TL in fathers from high SES and mothers and their newborns from low SES is in accordance with another study [6], which can be evidence for an increase in the probability of developing an illness and earlier diseases in both parents and their offspring [42]. Therefore, this influence of SES on parental health may cause DNA activity and remodeling or repair of genetic material and TL during fetal life [6,21]. The loss can also interfere with epigenetic regulation and may be due to the stresses involved in both high and low SES [12]. Regarding occupation, the majority of females in this study were not employed (homemakers) and had longer TL, which is opposite to other research results that showed longer telomeres of retired working women than homemakers (*p* = 0.023) [43]. Similar to the current study, research on working women indicated a decrease in 472 bp telomeres compared to nonemployed women [44]. The majority of fathers were working and had longer TL, which could be supported by a study of welders in Sweden showing the association (*p* = 0.033) of TL with oxidative stress in response to welding fumes. Exposure to fumes releases small peptides or other reactive species into blood, highlighting exposure-related changes in DNA methylation or telomere length due to alterations within cells, or blood cells due to infection or inflammation [45]. Similarly, in another study, fathers’ occupations showed a strong association with newborn telomere, depicting that it may have wear and tear effects on newborn health [46].

The current study did not encounter any significant ethnic-related differences from the TL study. Education was found associated with TL [21], which is in accordance with our previous study, emphasizes females educated up to graduation level have longer TL, as well as their newborns. Consistent with this study, many studies reinforced the hypothesis of a mother’s education and intellectual ability’s strong impact on the telomere attrition of their newborn [37,47]. So, females with less education during the perinatal period with lower values had shorter telomeres, due to less awareness of healthy living and laxness regarding distinct risk factors, thus leading to cellular aging and disease incidence. Hence, social status plays a crucial role in the transmission of telomere length in future generations.

The smokers included in this study had longer TL in males and their newborns than in females, highlighting the increase in TL can be due to an increased proliferative potential of the somatic cells by escalating telomerase enzyme activity, consequently causing cancer susceptibility, which is known to be caused due to the accumulation of de novo mutations that result in longer telomeres [48].

A study confirmed 4.2% shorter telomere (T/S ratio) with a positive association in multiparous females [49], which is similar to this study’s outcome of a decreased TL pattern with an increase in parity. Therefore, primigravida (female with first pregnancy) had been found affiliated with decreased risk of morbidity and mortality due to less exposure to the risk factors involved in pregnancies, leading to stable telomere biology [50].

When newborn parameters were evaluated, a comparison between girls and boys was seen. The mode of delivery showed surprising results. It was found that LSCS showed smaller TL (3.21 ± 3.51, 2.42 ± 2.90) (0.04) than SVD. The literature search revealed similar studies identifying longer TL in newborns with normal delivery highlighting SVD with an increase in immune cells (neutrophils, leukocytes, and NK cells), whereas LSCS is a stress-associated delivery with fewer circulating T cells and high Interleukin-7, thus emphasizing that such babies could be more prone to neonatal infectious diseases due to changes in the immune system and a decrease in the protective effect of telomeres [51,52]. This study also found an increase in TL in newborns with an increase in gestational age; however, another study reported a 25% decrease in placental RTL during the third trimester of gestation [53]. The reason for the increase in TL can be due to telomerase activation during cellular division with an increase in gestational age. Previous research on low birth weight infants found 240 bp per week telomere deletion during pregnancy, which may undergo a progressive deterioration causing senescence and early aging [54]. The increase in gestational age with an increase in birth weight can lead to TL fetal programming, which can also cause metabolic disturbance and modified tissue development. In the current research, newborn girls had longer TL than newborn boys, which is similar to many other studies (0.181 kb, 6.83%, 50–100 bp) [50,51]. However, longer TL is seen in girls due to the protective effect of hormones from reactive oxygen species [21,54].

This study had some limitations. First, because it was a cross-sectional study, we therefore worked on a limited sample size. Secondly, information on lifestyle and other social factors along with maternal nutrient insufficiency data were not collected. Finally, bias may exist in the self-reported variables. Telomeres serve as the biological timekeepers of cellular health, and hence, their attrition relative to chronological age is an indication of advanced biological aging, the existence of a stressed environment, and potential disease risk. Hence, improving the antenatal health of mothers and fathers by targeting modifiable factors through demographics can help to prevent in utero telomere attrition, enhance cellular longevity, and improve health outcomes. Future studies with a larger sample size should be conducted to validate the current findings and also focus on effective studies to reveal the role of the TERT gene polymorphism and its role in disease development, repair mechanism, mRNA expression, and telomerase protein level detection in parents and their newborns.

## 5. Conclusions

This study provided definitive evidence for the genetic influence (TERT gene) on triad telomere length (TL). TERT gene polymorphism highlighted A as a minor allele with shorter TL and the CC genotype had a positive association with newborn TL. Moreover, parental TL was found associated with newborn TL. The mother’s TL confirmed her antenatal health, immunosenescence, and clinical factors. However, all these are significantly associated with newborn TL, emphasizing the role of DNA repair, reprogramming, and terminal differentiation of T-cells (KLRG1, CD57), which may have transgenerational health effects.

## Figures and Tables

**Figure 1 cells-11-03777-f001:**
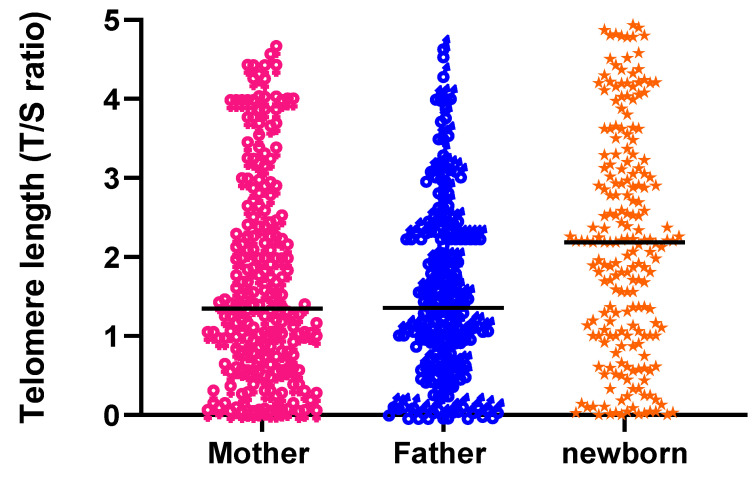
Telomere length (T/S ratio) comparison between mother, father and newborn. Newborn TL was longer, followed by mother and father TL.

**Figure 2 cells-11-03777-f002:**
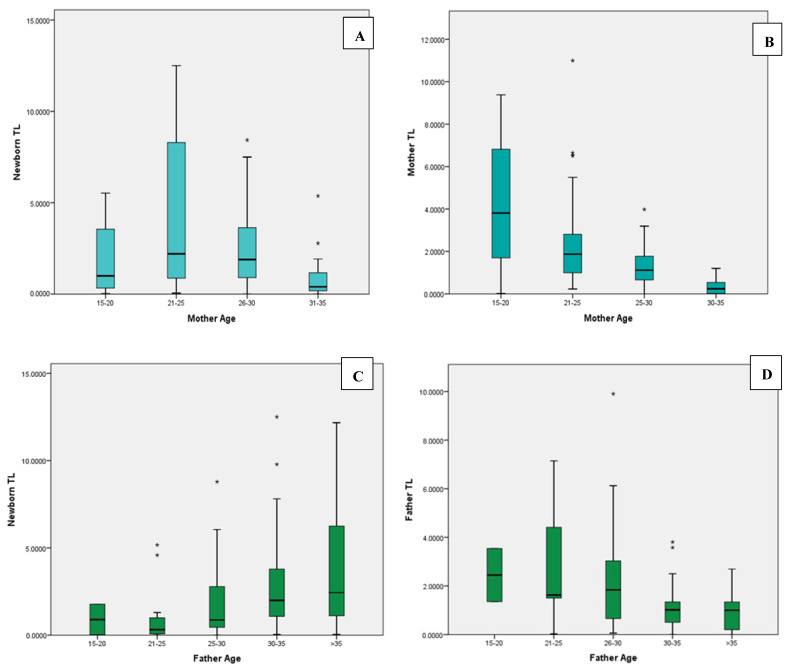
(**A**,**B**) Comparison of mother age with newborn telomere length. The longest newborn telomere length (TL) was seen in mothers aged 21–25, whereas TL in mothers decreases with an increase in their age. (**C**,**D**) Comparison of fathers’ age with newborn telomere length. An increase in fathers’ age showed an increase in newborns TL. The longest father TL was seen in ages 21–25. (*) symbol for *p* < 0.05; (**) for *p* < 0.01; (***) for *p* < 0.001.

**Figure 3 cells-11-03777-f003:**
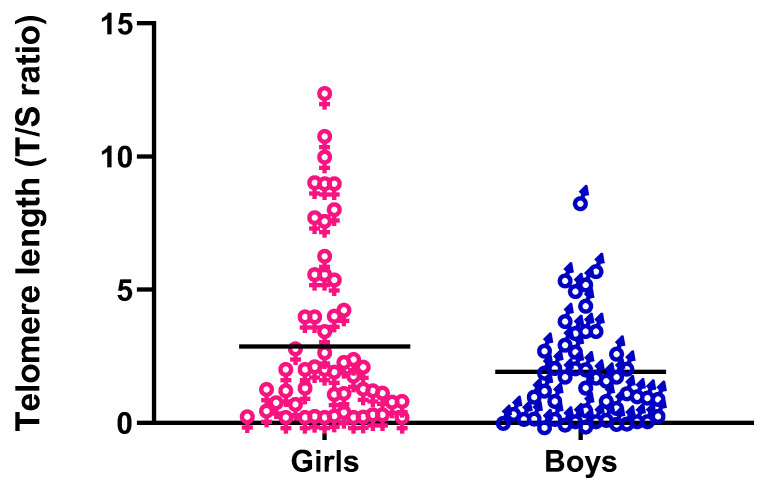
Telomere length (TL) comparison between newborn girls and boys. Girls have longer TL than boys.

**Figure 4 cells-11-03777-f004:**
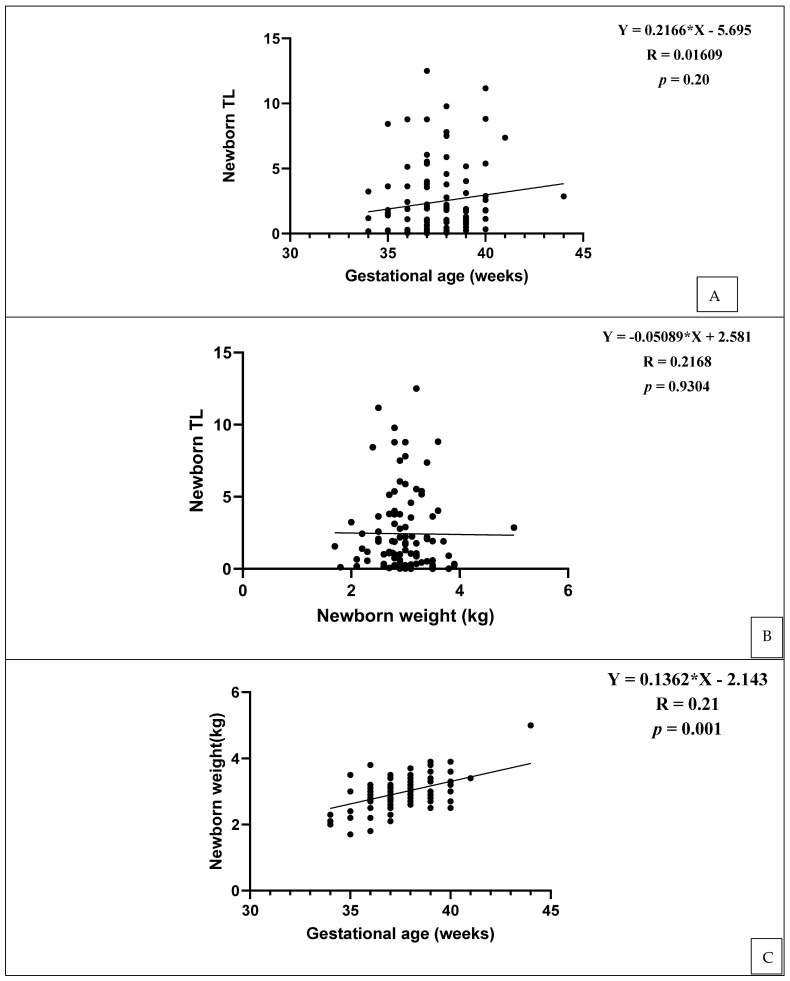
(**A**–**C**) Association between newborn telomere length (TL) and gestational age and birth weight. There was no association between newborn weight and gestational age, but there was a positive association between gestational age and newborn weight.

**Figure 5 cells-11-03777-f005:**
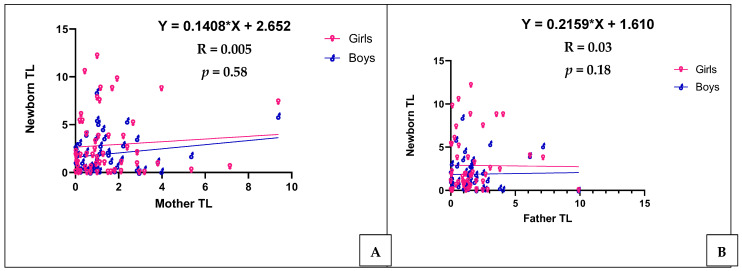
(**A**,**B**) Linear regression between mother and father TL among genders. No association was found between girls and boys with both parents.

**Figure 6 cells-11-03777-f006:**
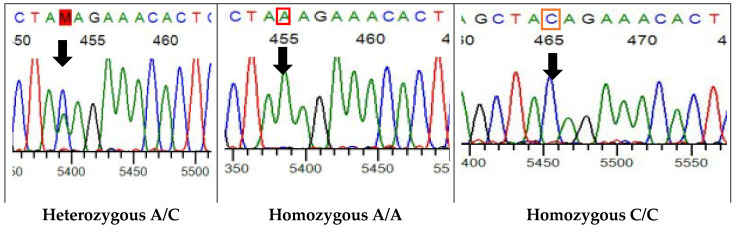
Sanger sequencing of TERT gene. Heterozygous genotype (AC) in both mothers and newborns but homozygous (AA, CC) in the father and newborns. The black arrow shows the SNP C/A allele at the restriction site rs2736100.

**Figure 7 cells-11-03777-f007:**
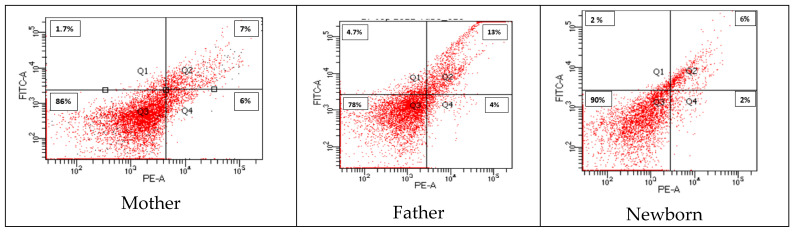
Comparison of immune markers (CD57, KLRG1) among mother, father, and newborn.

**Figure 8 cells-11-03777-f008:**
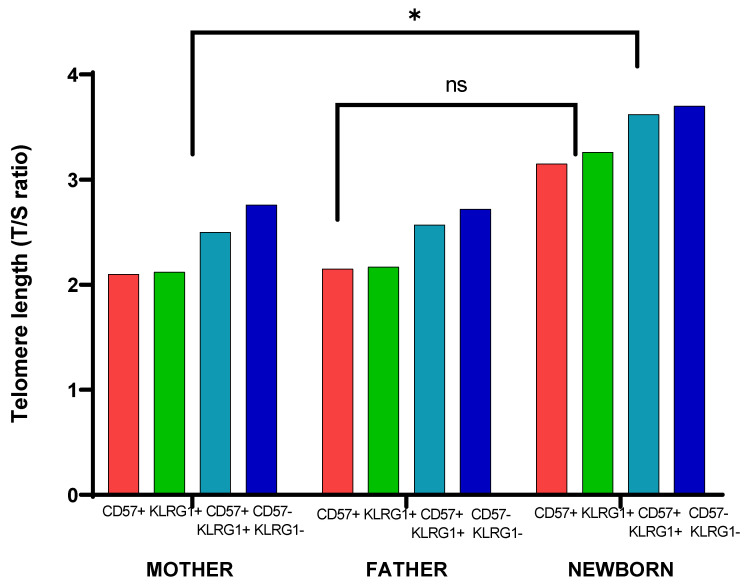
Comparison between telomere length and immune markers (CD57, KLRG1) in the triad: significant results (* *p* < 0.05) among mothers and newborns, whereas insignificant (*p* > 0.05) results between fathers and newborns. ns: not significant.

**Table 1 cells-11-03777-t001:** Comparison between mother, father, and newborn telomere length among different variables.

Variables *N* = 104	Mother *n* (%)	T/S Ratio Mean ± SD	*p*-Value	Father *n* (%)	T/S Ratio Mean ± SD	*p*-Value	Newborn T/S Ratio Mean ± SD	*p*-Value
Age (Mean ± SD)	27 ± 3.94	1.6 ± 2.00	0.000 ***	34.54 ± 6.01	1.49 ± 1.63	0.07	2.51 ± 2.87	0.003 ***
Socioeconomic Status								
Low	51 (25)	1.38 ± 1.14	0.02 *	51 (25)	1.83 ± 1.84	0.04 *	2.28 ± 2.01	0.015 **
Lower middle	51 (25)	1.91 ± 2.4		51 (25)	1.81 ± 1.96		2.36 ± 2.80	
Upper middle	51 (25)	1.99 ± 2.71		51 (25)	1.29 ± 1.48		2.58 ± 3.86	
High	51 (25)	1.52 ± 1.73		51 (25)	0.97 ± 0.974		2.83 ± 3.07	
Occupation								
Nonworking	164 (80.4)	1.70 ± 2.03	0.059	5 (2.5)	1.26 ± 0.70	0.086	2.23 ± 2.3	0.034 *
Working	40 (19.6)	1.40 ± 1.74		199 (97.5)	1.49 ± 1.64		3.2 ± 2.95	
Education								
No schooling	25(12.3)	1.64 ± 1.04	0.03 *	29 (14.2)	1.04 ± 0.43	0.067	1.62 ± 1.83	0.04 *
Less than high school	61 (33.8)	1.79 ± 1.74		45 (22.1)	1.94 ± 1.80		1.95 ± 1.29	
High school	37 (18.1)	2.15 ± 1.79		42 (20.6)	1.70 ± 1.83		2.27 ± 1.74	
Graduation	66 (32.4)	2.17 ± 1.67		66 (32.4)	1.90 ± 1.97		2.78 ± 2.89	
Masters	17 (8.3)	1.85 ± 1.88		22 (10.8)	1.80 ± 0.52		2.25 ± 1.60	
Ethnicity								
Pathan	87 (42.6)	1.89 ± 2.28	0.38	80 (39.2)	1.72 ± 2.03	0.29	2.73 ± 3.02	0.71
Urdu speaking (muhajir)	53 (26)	1.35 ± 1.59		45 (22)	1.20 ± 1.30		2.04 ± 2.56	
Punjabi	10 (4.9)	2.83 ± 2.80		15 (7.3)	1.89 ± 2.49		1.63 ± 1.89	
SARIKI	9 (4.4)	1.47 ± 1.91		4 (1.9)	1.03 ± 0.54		1.14 ± 1.55	
Sindhi	27 (13.2)	2.07 ± 1.56		47 (23)	1.15 ± 0.68		2.60 ± 2.71	
Other	18 (8.9)	1.20 ± 1.47		13 (6.3)	1.19 ± 1.02		1.31 ± 1.51	
Smoking status								
Smoker	14 (6.8)	1.76 ± 1.56	0.24	60 (29.4)	2.13 ± 2.08	0.046	2.51 ± 2.99	0.025 *
Nonsmoker	190 (93.2)	1.64 ± 1.61		144 (70.6)	1.28 ± 2.02		2.08 ± 1.98	

(*) symbol for *p* < 0.05; (**) for *p* < 0.01; (***) for *p* < 0.001.

**Table 2 cells-11-03777-t002:** Maternal antenatal clinical parameters and their effect on newborn telomere length.

Parameters	Mother *n* (%)	T/S Ratio (Mean ± SD)	*p*-Value	Newborn T/S Ratio (Mean ± SD)	*p*-Value
Hemoglobin (Hb) mg/dL					
<8–10	44 (26.5)	1.33 ± 1.46	0.05	1.96 ± 1.81	0.02 *
10.1–11.9	112 (60)	1.72 ± 1.79		2.80 ± 3.09	
>12	38 (18)	1.94 ± 3.03		2.93 ± 3.07	
Total leukocyte count (TLC) 10^9^/L					
6.1–8	38 (18.7)	1.48 ± 1.47	0.09	2.71 ± 3.72	0.06
8.1–10	72 (35.3)	1.77 ± 2.22		1.78 ± 1.72	
10.1–12	59 (28.9)	1.31 ± 1.45		2.98 ± 2.89	
>12	35 (17.2)	1.92 ± 2.78		2.99 ± 3.55	
Parity					
0	59 (28.9)	2.09 ± 2.29	0.09	2.75 ± 2.16	0.025 *
1	58 (28.4)	2.04 ± 2.41		2.50 ± 2.75	
2	44 (21.6)	1.64 ± 1.71		2.17 ± 3.64	
3	25 (12.3)	1.34 ± 1.72		1.98 ± 1.41	
4 or >4	18 (4)	1.07 ± 1.03		1.80 ± 3.12	
Gravidity					
1	58 (28.5)	1.84 ± 1.79	0.061	3.68 ± 2.50	0.05
2	54 (26.5)	1.71 ± 2.50		3.59 ± 2.34	
3	36 (17.6)	1.70 ± 1.62		2.17 ± 2.78	
4 or >4	56 (24)	1.37 ± 1.76		1.88 ± 2.17	
Miscarriages					
No	134 (65.2)	1.45 ± 1.85	0.78	2.47 ± 2.96	0.45
Yes	71 (34.8)	1.34 ± 0.95		2.82 ± 2.52	

(*) symbol for *p* < 0.05.

**Table 3 cells-11-03777-t003:** Difference between telomere length (TL) among newborn gender.

Parameters		Girls *n* (%) 54 (53)	T/S Ratio Mean (SD)	Boys *n* (%) 50 (47)	T/S Ratio Mean ± SD	*p*-Value
Mode of Delivery (MOD)	SVD	23 (43)	3.21 ± 3.51	23 (46)	2.01 ± 1.65	0.04 *
	LSCS	30 (56)	2.61 ± 3.06	27 (54)	2.24 ± 2.90	
Gestational age (weeks)	36–38	41 (77)	2.50 ± 2.93	39 (78)	2.24 ± 2.61	0.20
	39–42	12 (23)	3.8 ± 3.82	11 (22)	2.69 ± 1.55	
Birth weight (kilogram)	2–2.5	6 (13)	3.01 ± 4.19	8 (15)	2.50 ± 2.83	0.93
	2.6–3.0	31 (61)	3.03 ± 3.15	26 (51)	2.68 ± 1.62	
	3.1–3.5	16 (26)	3.60 ± 2.31	16 (34)	2.83 ± 3.17	
Heart Rate (HR)	130–140	30 (55)	2.90 ± 3.13	28 (56)	2.16 ± 2.88	0.03 *
	141–150	24 (45)	2.14 ± 2.89	22 (44)	2.06 ± 1.93	
Respiratory Rate (RR)	41–50	39 (72)	2.72 ± 2.92	30 (60)	1.97 ± 2.37	0. 01 **
	51–60	15 (28)	3.68 ± 4.08	20 (40)	1.74 ± 2.03	
Length/height (cm)	30–40	02 (4)	3.47 ± 0.97	-	-	0.04 *
	40–50	38 (70)	2.94 ± 3.29	35 (70)	1.92 ± 1.95	
	51–60	14 (26)	2.42 ± 3.37	15 (30)	3.62 ± 3.21	
Occipital fronto circumference (OFC) (cm)	30–35	48 (89)	2.87 ± 3.35	50 (100)	2.1 ± 2.39	0.01 *
	36–40	06 (11)	2.85 ± 2.44	-	-	
Mean telomere length (T/S)		2.72 ± 3.09		2.13 ± 2.39		0.000 ***

*p* < 0.05; the (*) symbol for *p* < 0.05; (**) for *p* < 0.01; and (***) for *p* < 0.001.

**Table 4 cells-11-03777-t004:** Comparison between the genotype and telomere (TL) among mother, father, and newborn (triad).

Variables	Mother *n* = 32 (32.6)			Father *n* = 32 (32.6)			Newborn *n* = 32 (32.6)		
TERT Genotype	A/C	A/A	C/C	A/C	A/A	C/C	A/C	A/A	C/C
Frequency	22 (0.687)	0	10 (0.312)	7 (0.218)	6 (0.187)	19 (0.593)	24 (0.705)	8 (0.235)	2 (0.058)
TL	1.35 ± 1.21	-	1.28 ± 0.36	1.40 ± 0.13	0.60 ± 0.77	1.13 ± 0.30	3.47 ± 3.7	-	1.50 ± 1.40
β0 (95% Cl)	1.19 (0.8175 to 1.662)			1.028 (0.7729 to 1.283)			2.19 (0.9951 to 3.386)		
SE	0.17			0.12			0.57		
*p*-Value	<0.0001			<0.0001			<0.0011		

A/C: heterozygous allele; A/A, C/C: homozygous allele; SE: standard error.

**Table 5 cells-11-03777-t005:** Major and minor alleles effect on Telomere Length in different populations.

Country	Major Allele	Minor/Alternative Allele	Telomere Length	*p*-Value	References
Pakistan	C	A	Shorter	<0.0001	Current study
China	A	T	Shorter	0.046	Weng et al. 2016 [27]
	T	C	Longer	1.93 × 10^−5^	Liu et al. 2014 [29]
United Kingdom	C	A	Shorter	0.986	Michalek et al. 2017 [30]
Asia	C	A	Low telomerase	0.018	AlDehaini et al. 2021 [31]

## Data Availability

The data presented in this study are available on request from the corresponding author.

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
