# Peer review of "Parental Genetics Communicate with Intrauterine Environment to Reprogram Newborn Telomeres and Immunity"

_cells, 2022, doi:10.3390/cells11233777_

Round 1

Reviewer 1 Report (Previous Reviewer 2)

The manuscript has been improved.

Reviewer 2 Report (Previous Reviewer 3)

Revisions appear to adequately address the prior critiques in the revised manuscript by Farrukh et al.

This manuscript is a resubmission of an earlier submission. The following is a list of the peer review reports and author responses from that submission.

Round 1

Reviewer 1 Report

The manuscript titled ‘Parental genes communicate with intrauterine environment to 2 reprogram newborn telomeres’ by Farrukh describes an experiment that measured the telomere lengths (TL) of newborn babies as well as their parents, aiming to identify factors that affect offspring’s TL. They claimed to have identified six new factors that affect TL in newborn. However, these factors have been reported before to affect TL, thus, the finding lacks novelty. My main criticism is that the sample number is too small for a populational study, particularly with multiple factors analyzed, so that there are only limited numbers in each group.  In addition, it is difficult to understand their interpretation of some of the data. For example, 1) how could the positive association between parents and newborns emphasizes ‘fetal programming or repair of the telomere biology’ (page 13, line 295)? 2) ‘Maternal and newborn TL was significantly associated with maternal age ranging from 18 to 35 years’ – please clarify what association it is. 3) Newborn TL ‘showed no association with paternal age’ but ‘newborns of older fathers had longer TL in the analysis’ – are they contradictory?

In summary, I feel that the manuscript is not well written and it is difficult to follow the logic of authors on their experimental design and interpretation.

Author Response

  1. Yes we claimed to have identified six new factors that affect TL in newborns. These factors have been reported before to affect newbornTL either in the mother or father. our work is novel because it is the first time data presented in our targeted population and in literature such collective work on triads is scarce. 
  2. the sample number is small because of logistic issues and small grants of Rs. 1.6 million only. we also used targeted population word rather than referring it as a population-based study.
  3. The positive association between parents and newborns emphasizes ‘fetal programming or repair of the telomere biology’ (page 13, line 295) because at the start of the paragraph we highlighted that newborns had longer TL than parents and if it is positively associated that mean it is confirming the line 293 investigation.
  4. ‘Maternal and newborn TL was significantly associated with maternal age ranging from 18 to 35 years’ – please clarify what association it is. It is negatively associated 
  5. 3) Newborn TL ‘showed no association with paternal age’ but ‘newborns of older fathers had longer TL in the analysis’ – are they contradictory? we had mentioned that it was the study finding or observation but had no association.

Reviewer 2 Report

The manuscript describes a study on parental TL was found associated with newborn TL.

It is easily that newborn TL was longer followed by mother and father TL due to age different.

But why is different between newborn gender girls and boys ?

Author Response

It is easily that newborn TL was longer followed by mother and father TL due to age different.

Ans. the age of mothers showed different pattern of TL than fathers.

But why is different between newborn gender girls and boys ?

Ans. It is found in literature and our previous study that girls have longer TL because of the protective effect of hormones from reactive oxygen species and telomere than boys. we have added to the discussion

Reviewer 3 Report

The manuscript by Farrukh et al reports on the effects of parental TL, Tert variant, and prenatal uterine health on newborn TL. The effect of parental TL on newborn TL is already established in the literature, and the additional data on effects of Tert variant and prenatal uterine health on newborn TL do not warrant publication in Cells. This manuscript would be a better fit in a more specialized journal.

Author Response

The effect of parental TL on newborn TL is already established in the literature.

The literature is scarce on the work in the triad and this is the novelty of work because it is highlighting many factors together.

 effects of Tert variant and prenatal uterine health on newborn TL do not warrant publication in Cells. 

TL is always related to genetics and that is the small portion of the article. 

Declining on the novelty and above-mentioned comments is not encouraging. 

Journal had invited for publication in the special issue and should reconsider the article with changes.